# Skin Cortisol and Acoustic Activity: Potential Tools to Evaluate Stress and Welfare in Captive Cetaceans

**DOI:** 10.3390/ani13091521

**Published:** 2023-05-01

**Authors:** Chung-Hei Wong, Ming-An Tsai, Fung-Chi Ko, Jiann-Hsiung Wang, Yi-Jing Xue, Wei-Cheng Yang

**Affiliations:** 1School of Veterinary Medicine, National Taiwan University, Taipei 106, Taiwan; 2Collage of Veterinary Medicine, National Pingtung University of Science and Technology, Pingtung 912, Taiwan; 3National Museum of Marine Biology and Aquarium, Pingtung 944, Taiwan; 4Institute of Marine Biology, National Dong-Hwa University, Pingtung 944, Taiwan; 5College of Veterinary Medicine, National Chiayi University, Chiayi 600, Taiwan

**Keywords:** beluga whale, *Delphinapterus leucas*, stress, cortisol, acoustic activity

## Abstract

**Simple Summary:**

Animal welfare is a main concern for captive animal keepers. The aim of this study was to evaluate the non-invasive vocalizations and sloughed skin scrape cortisol concentrations that are associated with potential stressful contexts in order identify the sources of stress in captive beluga whales. This study validated the use of beluga sloughed skin scrape as a matrix for measuring cortisol, measured the cortisol concentration using routinely collected samples, and identified the potential events (stressors) by examining medical and husbandry records. Besides, we estimated the time lag between the dates of the event and the peak in the cortisol concentration (response) in the skin samples, and analyzed the changes in the acoustic activities of the beluga whales during the selected events. The findings of this study may offer constructive recommendations regarding the environmental or internal stressors that captive cetaceans experience, and contribute to developing strategies to reduce or eliminate the stressors.

**Abstract:**

As people’s focus broadens from animals on farms to zoos and aquaria, the field of welfare science and the public’s concern for animal welfare continue to grow. In captive animals, stress and its causes are topics of interest in welfare issues, and the identification of an objective method that can be used to assess animals’ stress as a physiological state is essential. Both behavioral and physiological parameters can be used as indicators in order to assess animal stress quantitatively. To validate this approach, acoustic activity and the sloughed scrape skin cortisol concentration were used to evaluate the animal welfare of captive beluga whales (*Delphinapterus leucas*). The acoustic activity (5 min at 10:00 am) of three captive *D. leucas* was routinely recorded by a transducer and analyzed using audio editing software. The calls were separated into three main categories: whistles, pulses, and combo calls. The sloughed scrape skin samples were collected non-invasively once a week from all three animals’ fluke and/or flipper. Cortisol was extracted using a modified skin steroid extraction technique, and detected via commercially available enzyme immunoassays. The results showed that the cortisol concentration increased by varying levels when the whales encountered the same event. In addition, the number and distribution of the calls changed along with the events. This indicated that the changes in the cortisol concentration and acoustic behavior may have reflected the fluctuations in the environment and body condition. Therefore, the scrape cortisol measurement and acoustic recordings could be used to monitor stress levels in captive beluga whales. We recommend that aquaria consider incorporating skin scrape cortisol and acoustic activity monitoring into their standards for animal welfare.

## 1. Introduction

The Five Domains model, initially formulated in 1994 [1], is the fundamental form of animal welfare assessment [2]. The model consists of four physical domains, namely nutrition, physical environment, health, and behavioral interactions, and a fifth mental state domain. In captive animals, which are confined to a limited environment and to restricted social grouping in farms, zoos, laboratories, or personal ownership, stress and its causes are topics of interest in welfare issues [3]. The relationship between welfare and stress has frequently been studied in the last 40 years. Stress can cause pre-pathological development, which is the most credible measure of the stress affecting an animal’s well-being [4]. When the symptoms of stress appear and an individual loses control of their living conditions, it is possible that maintaining welfare becomes difficult [5]. A more expansive view suggests that all biological responses can reflect the state of welfare and that the development of various pathologies is a manifestation of excessive stress, which is also an indicator of a poor welfare status [6]. Stressors that disrupt the homeostasis are believed to cause measurable behavioral and physiological changes, which can sometimes impair the animal’s welfare [7]. Therefore, animal welfare is a main concern for captive animal keepers.

It is widely accepted that stress response plays an essential role in the overall health of animals. Stress response refers to a suite of physiological changes that occur in response to a threat or stressor. In many cases, the physiological stress response is an adaptive feature of natural stressors. Stress hormones such as cortisol, aldosterone, and epinephrine function by increasing the heart rate, blood pressure, and respiratory rate in order to more efficiently circulate energy and oxygen to tissues [8]. However, when stress is prolonged or triggered too often, it becomes chronic stress and can lead to compromised health and an increase in the captive animals’ susceptibility to diseases, which can lead to a poor welfare status [7,8,9,10]. Managing stress, and ensuring proper medical treatment and animal husbandry, are essential to captive animal care. Identifying stressors in the captive environment is crucial for managing stress in captive animals and thus facilitating good welfare. The measurement of glucocorticoids (GCs, cortisol and corticosterone) in the blood, feces, urine, saliva, tears or hair is commonly used to monitor and identify stress in zoo and wildlife medicine [11,12,13,14]. The adrenal gland secretes GCs via the activation of the hypothalamic–pituitary–adrenal (HPA) axis and exerts a variety of physiological and behavioral effects [15]. The use of the concentrations of these GCs to indicate the overall physiological state and provide vital information on the health and resilience of a population has been proposed [16]. Although GCs are commonly used markers for evaluating stress in animals [15,17,18,19,20,21], the assumption that the activation of the HPA axis and increase in GCs are the result of negative stimuli or stressors is not always correct [22]. This is because the HPA axis is also involved in energy mobilization and redistribution [23]. In addition, it is believed that no single biological parameter can adequately inform on a stressful condition and that no single stress response is present in all stress-related situations [7]. Using multiple measures to identify stressors could enhance the scientific validity of GCs as indicators of welfare.

Despite the fact that cetaceans have been maintained in a captive environment for over 150 years, limited scientific research has been dedicated to their welfare [24]. Indeed, there are only several established methods used to quantify and address the stress and welfare of captive cetaceans [24,25]. To the best of our knowledge, published empirical articles that focus on captive cetacean stress and welfare are limited to four studies on bottlenose dolphins (*Tursiops truncatus*) [3,18,26,27] and one evaluation of beluga whales (*Delphinapterus leucas*) [28]. One of the four *T. truncatus* studies used a single physiological parameter, the salivary cortisol concentration, to analyze the stress response after construction activities [18], while the other three *T. truncatus* studies used behavioral and physiological parameters to study the stress response. It was suggested that the behavioral (observation) record might be an important early indicator of health problems after investigating the correlation between behavioral indices and physiological measures (including basic blood parameters and postmortem findings) of stress and health [3]. One study found that behavior and the salivary cortisol concentration varied among dolphins in open and closed facilities, suggesting that the design of pools may influence welfare [27]. Furthermore, the willingness to participate (WTP) was suggested to be a potential behavioral measure and an indicator of early changes in dolphins’ health state; this finding was obtained when comparing the dolphins’ WTP in the training sessions with the dolphins’ health state (health and appetite), and finding that the WTP decreased significantly when the dolphin was sick [26].

Vocal behaviors have been used as an indicator of stress in captive terrestrial mammals (e.g., cattle: branding; domestic pigs: castration; guinea pigs: isolation) (reviewed in [29]). Because cetaceans have complex acoustic repertoires, it is thought that the vocal behavior of cetaceans could be used to monitor their stress. Beluga whales have high acoustic abilities and rich vocal repertoires, earning them the nickname the ‘marine sea canary’ [30]. Their vocalization behaviors have been studied in captive and free-range environments [31,32]. In addition, their calls have been widely described and classified [33,34,35]. Previous studies have suggested that there are several similarities in the call types of cetaceans and in the presence of novel call types among different populations, suggesting that there are differences in the vocal behavior of cetaceans among various geographic regions [36]. Furthermore, the acoustic activities of cetaceans have been supposed to be a more accurate method of evaluating the effect of environmental stressors in beluga whales than behavioral observation [28].

Vocalization parameters alone, such as the vocal rate, intensity, and frequency range, have been found to vary with stress in bottlenose dolphins [3]. However, some studies have simultaneously examined both physiological and behavioral parameters in an effort to quantitatively evaluate physical well-being (reviewed in [7]). In a previous study [37], the researchers compared the changes in the cortisol concentration and cytokine gene transcription level in blood samples and the behaviors of captive bottlenose dolphins after the exposure to sound. The study showed that the sound stimuli was a stressor for cetaceans, as significantly increased cortisol levels and a stress-induced cytokine shift were noticed, although only minor behavior changes were observed. The potential acute or chronic stress that is ignored by traditional observation could lead to compromised health and an increase in captive animals’ susceptibility to diseases and to a poor welfare status. Another study on captive beluga whales [38] showed that different cytokine transcript profiles were found in different clinical problems (open wound, skin lesions, social stressors, and an abnormal blood iron level). This indicated that no single biological parameter can adequately inform on a stressful condition and that no single stress response is present in all stress-related situations. The more that sophisticated psychophysiological parameter measures are employed, the more valid the interpreted results are. Therefore, there is a need for a precise and effective method to measure the physical well-being of captive cetaceans. The aim of this study was to evaluate the non-invasive vocalizations and sloughed skin scrape cortisol concentrations that are associated with potential stressful contexts in order identify the sources of stress in captive beluga whales. The objectives included (i) validating the use of beluga sloughed skin scrape as a matrix for measuring cortisol; (ii) measuring the cortisol concentration using routinely collected samples; (iii) identifying the potential events (stressors) by examining medical and husbandry records; (iv) estimating the time lag between the dates of the event and the peak in the cortisol concentration (response) in the skin samples; and (v) analyzing the changes in the acoustic activities of the beluga whales during the selected events. The findings of this study may offer constructive recommendations regarding the environmental or internal stressors that captive cetaceans experience, and contribute to developing strategies to reduce or eliminate the stressors.

## 2. Materials and Methods

### 2.1. Scrape Samples Collection

Three beluga whales were recruited for this study, including an adult female (A) and two adult males (B and C). These animals were captured off the shore of Russia at three years of age in 2002 and kept in a local aquarium. The following trade and transportation proceeded by following the laws and regulations of the Taiwanese government. They were then housed in indoor pools with 17 to 18 °C filtered and ozone-sterilized natural seawater at the National Museum of Marine Biology and Aquarium (NMMBA) in Pingtung, Taiwan. The facility’s lighting was controlled to provide ten hours of daylight daily. The beluga whales’ husbandry and medical records since January 2017 were collected, including their daily food intake, behavior, environmental enrichment, training process, physical and health assessment, treatment when health problems occurred, and any environmental changes, such as the weather, water temperature, the depth of the water, construction and maintenance, and other unusual items, such as visitors and noise, which were controlled in compliance with the laws and regulations of Taiwan’s government regarding captive wildlife animals.

Skin scrape samples, which comprise the outermost part of the skin and mainly consist of epidermal cells with a high water content (Figure 1), were collected from these three animals once a week from March 2017. The protocol was reviewed and approved by the IACUC of NMMBA and Ocean Affairs Council (approval number 1070003656). The beluga whales that participated in this study had received sampling training to prevent potential stress during sample collection. In the early stage of the experiment, it was observed that collecting samples from the fluke and flipper could provide enough samples, but the body could not. Therefore, the animals were also trained to show their flippers and flukes voluntarily. A sterilized disposable wooden tongue depressor was run along the animals’ left flipper or fluke multiple times to collect the epidermal cells in a tube. The sampling location for each animal was the same. In the routine monitoring, the sample from beluga A was collected from the dorsal or ventral side of the fluke, and the samples from beluga B and C were collected from the flipper (Figure 2). In the flipper–fluke comparison experiment, seven sets of scrape samples from the flipper and fluke were collected on the same day in order to analyze the variations among different sampling locations. After collecting, the samples were immediately centrifuged at 10,000 rcf for 3 min, and then the supernatant was removed.

### 2.2. Hormone Extraction

The method of extracting the steroid hormones from the skin, as described here, is a modified form of the process described in the work of Kellar et al. [39]. Glass beads for homogenization and the grinding media (glass beads) remained in the tube when adding the ethanol/acetone solvent. Approximately 0.08–0.15 g of scrape sample was homogenized by vortexing it with 1000 μL ethanol and 0.36 g of 2 mm glass beads for 30 min. Then, the mixture was centrifuged at 3000 rcf for 10 min. The homogenate/glass beads solution was combined with 2 mL of 4:1 ethanol/acetone and vortexed for 10 min and centrifuged at 5000 rpm for 10 min. The supernatant was transferred into a new tube and evaporated by a stream of N2 vapor. Next, 2 mL of diethyl ether was added to the evaporated contents, vortexed, and centrifuged at 3000 rcf for 15 min. The supernatant was collected and evaporated, and the residue was resuspended in 1500 μL of acetonitrile, vortexed, and 1500 μL of hexane was added to the mixture. After the solution was vortexed and centrifuged again, the hexane layer was removed, and the process was repeated with another 1500 μL of hexane. The final acetonitrile layer was collected and evaporated. The final residue was centrifuged at 2500 rcf rpm for 5 min and stored at −20 °C.

### 2.3. Cortisol Detection

To prepare the samples for the enzyme immunoassay (EIA), the residue was suspended in 150 μL of 1 M phosphate-buffered saline and then was vortexed for 15 min. A commercial enzyme immunoassay was used to quantify the cortisol concentrations (#K003-H1, Arbor Assays, Ann Arbor, MI, USA), which had been used in other cetacean studies [39,40]. The manufacturer’s protocols were followed exactly for the detection of cortisol.

The extraction efficiency, using a spiked sample, was determined by following the protocol described by Kellar et al. [39]; as such, 320 pg of cortisol was spiked with a known cortisol concentration sample. The extraction efficiency was calculated by using the following equation: E = Conc_0_ − Conc_K_/320, where E is the extraction efficiency, Conc_0_ is the sample concentration from the EIA kit (pg/mL), and Conc_K_ is the concentration of the known sample. To calculate the final hormone concentration of each sample, the following equation was used: Conc_F_ = (Conc_1_ × V_PBS_)/(1000 × M × E), where Conc_F_ and Conc_1_ are the calibrated concentrations of each sample(ng/g) and the sample obtained from the EIA kit (pg/mL), respectively, V_PBS_ is the volume of PBS (mL) used to suspend the sample, M is the wet weight (g) of each sample, and E is the efficiency.

### 2.4. Parallelism and Matrix Effects Analyses

Two additional quality control assessments were performed to gauge the performance of using the scraped extracts with the cortisol EIA kit. Parallelism was tested via an assay of six serial dilutions of scraped extract run along with known doses cortisol standards in order to determine whether the linear decrease in the measured values of the scraped extracts was parallel to the standard curve; this is an indication that the assay is measuring the same antigens in the scrape as in the standards. A 2-fold serial dilution comprised the sample concentration. Each dilution was run once, and the resulting curve of the detection metric (optical density of the sample/optical density when no sample is added(B/B0)) as a function of the dilution state was then compared to the standard curve using an F-test.

The second assessment examined the potential matrix interference, which is the effect of the scrape extract on the measurement of cortisol. A standard solution was spiked with phosphate-buffered saline and/or a set of serial dilutions of a pooled sample composed of sixteen beluga scrape cortisol extracts, which made a final equivalent volume of 150 μL. Each of the four serial dilutions was run in duplicate. The concentration of cortisol contributed from the pooled sample was subtracted from each sample-spiked measurement so that its contribution would be factored out of the assessment.

### 2.5. Time Lag Estimation

The signals of cortisol and other hormones are believed to be due to skin renewal, which can deliver hormones to the margin of the skin [41]. Therefore, there would be a time lag between the date of the event and the presence of a cortisol signal in the scrape. It was assumed that the time lag would be either 28–32, 38–42, 48–52, 58–62, 68–72 or 78–82 days. The cortisol concentration data and the husbandry/medical records were compared in order to find the most plausible time lag. An event was defined as a recorded unusual thing that had happened, but had not been confirmed as causing a stress response. A peak in the cortisol concentration (response) was defined as a value that was higher than the third quartile of the data obtained from one particular whale. Outliers were defined as values higher than the third quartile + 1.5 × IQR or less than the first quartile − 1.5 × IQR, where IQR is the interquartile range.

The date of the event was analyzed alongside the date of the response in each whale in order estimate time lags. One point would be given when the D_P_ (presumed date of response) matched the D_R_ (the date of response), where D_P_ = D_E_ (the date of event) + TL (time lag). When the event happened more than once, and a response was observed only once, that response could not get a point. The points from each whale were pooled as the total points of a certain time lag. The time lag that had the most points was then applied.

### 2.6. Acoustic Activities and Analyses

In order to record the acoustic activities of the three whales, the transducer (SM4 with hydrophone, Wildlife Acoustics, MA, USA) was placed next to the gate (Figure 3), which was kept enclosed to protect the transducer from damage caused by the whales. A 5 min recording at 10:00 am, which was neither a feeding session, an interaction with the keepers, nor within half an hour of any husbandry procedure, was regarded as the representative of each day. The acoustic activities were recorded at a sampling rate of 16 bit/96 kHz from March 2017 to March 2018. Several events were selected from the analyzed events in order to identify the time lag based on the acoustic recording quality. The 5 min daily acoustic activities of the day of the event, and those two days before and after the event, were analyzed both aurally and visually using sound analysis software (Kaleidoscope Pro, Wildlife Acoustics) with an FFT size of 2048 and a WIN size of 1024.

According to the previously published studies, the calls were separated into three main categories: (1) whistles, (2) pulses, and (3) combo calls. This was based on the acoustic parameters and variations in the vocalization spectral contours [31,33,35]. The calls that did not belong to the three main categories were categorized as unknown. The total number of calls and the pattern of the calls were identified by two experienced observers.

### 2.7. Statistical Analysis

An F-test was used to assess the differences between the slope of the standard solution and the pooled scrape extracts. Variations in the sampling location (flipper vs. fluke) were analyzed by linear regression. Descriptive statistics, an analysis of variance and the Kruskal–Wallis test were applied to the cortisol samples. A chi-squared test of homogeneity was used to analyze the variation in an event’s acoustic activities. Qualitative acoustic activities data were further analyzed using a principal component analysis (PCA).

## 3. Result

### 3.1. EIA Performance

The estimated extraction efficiencies for the skin samples (41.6% ± 15.9%) were based on recovering 320 pg of 15 cortisol spiked extracts. These measured extraction efficiencies were used as a correction factor, which was then applied to all blubber and scrape cortisol measurements within this study. The EIA standards and the pooled scrape extracts exhibited statistical parallelism when analyzed via an F-test. No significant differences (*p* > 0.05, Figure 4) were found in the slope between the linear portions of the binding curves of the serially diluted scrape extracts when compared to the pure hormone standards used in the same assay. No obvious trend was found in the anticipated concentration when the standard solution was spiked with increasing amounts of blubber extract (r^2^ = 0.0537 118.8 ± 11.2 pg/mL); this finding is consistent with there being little to no evidence of matrix interference.

### 3.2. Scrape Cortisol

The difference between the two sampling locations (flipper and fluke) showed that the cortisol concentrations from these two locations were similar, and the regression showed that the model fit the data well (Figure 5, R^2^ = 0.9685). Thus, the cortisol concentration of the samples from these two locations could be similar. The cortisol concentration of the scrape samples (*n* = 92) from the three whales ranged from 0.213 ng/g to 8.55 ng/g (wet weight, wt). The cortisol concentrations showed no significant difference among the three whales (Kruskal–Wallis test: H = 1.28, *p* > 0.05) (Figure 6). In beluga A, there was one outlier, and the CV was 67%. There were three and four outliers in beluga B and C, in which the CV was 113% and 125%, respectively. When the outliers were not included, there were still no differences in the cortisol concentration among the three whales (Kruskal–Wallis test: H = 1.19, *p* > 0.05), and the CV of beluga A, B and C were 54%, 63% and 41%, respectively.

The comparison between the response, events and time lag showed that days 68–72 had the highest points (Table 1), indicating that the scrape cortisol in beluga whales may present the stress response to events occurring around two months prior. Six responses to events were observed in beluga A, five in beluga B and four in beluga C. Therefore, 11 events were the potential cause of the stress responses observed in the beluga whales (Figure 7). The events corresponding to the responses are described in Table 2. Five events (event I, IV, VII, VIII and IX) with high-quality recording were selected for further acoustic activities analysis. The descriptions of the five events are as follows. In event I, maintenance was begun on 3rd February close to the pools and lasted for five days. According to the keepers, the drilling noise and vibration were noticeable only on the first day. Event IV was a case of enteritis experienced by beluga C. Event VII was an indoor pool gate repairment that was completed in one day, and only a metallic rapping noise was produced, with no vibrations. Event VIII was an annual routine fire drill that made sounds inside and outside of the aquarium. Event IX was an impulsive sound of unknown source from the sea embankment. During all event periods, except event IV (enteritis in beluga C), the food intakes were successful for these animals and no negative behaviors were observed.

### 3.3. Acoustic Activity

The dates of events I, IV, VII, VIII and IX were 3 February, 4 April, 6 June, 7 August and 17 September, respectively. The events were categorized into physiological disorder (event IV) (intrinsic) and environmental changes (extrinsic) (event I, VII, VIII and IX). In total, 2650 calls were identified from the recording of the five events. Among these, pulse calls (*n* = 1830), whistle calls (*n* = 497), combo calls (*n* = 311) and unknown (*n* = 12) were identified. The number of calls plummeted on the day of the event (Figure 8). The results of the chi-squared test of homogeneity (*p* < 0.05) and PCA (Figure 8) showed that the call distribution differed among the five days of the event. In events I, IV, and IX, pulse calls were in the majority on the day of the event compared to on Day 1 and 2, and the number of calls on Day 5 was still lower than the mean of those on Day 1 and 2. This showed another situation in event VII and VIII.

## 4. Discussion

Previous studies on cetacean cortisol levels have used serval different sample sources, which include blood (e.g., in *T. truncatus* [13,42] and *D. leucas* [43]), feces (e.g., in bottlenose dolphins [13,42] and right whales, *Eubalaena glacialis* [44]), blow (e.g., in *D. leucas* [45]), saliva (e.g., in bottlenose dolphins [18]), skin (e.g., in harbor porpoises, *Phocoena phocoena* [46], and bottlenose dolphins [41]) and blubber (e.g., in short-beaked common dolphins, *Delphinus delphis* [39], narwhals, *Monodon monoceros* [47], and beluga whales [43,48]). The concentration of cortisol in blood samples has been suggested to reflect the acute stress level [13,42,43], while skin, blubber and baleen have been suggested to reflect the cortisol status over multiple weeks and years [12,39,41,49]. Because the measurement of cortisol in the blood, saliva, and feces in live animals shows several disadvantages, such as frequent and invasive sampling and circadian variation, sloughed skin scrape has been suggested to be non-invasive biological sampling material for studying the stress load of cetaceans [41]. It is believed that the incorporation of cortisol (and other hormones) into the skin is due to skin renewal, which can deliver hormones to the margin of the skin [41]. However, the time between stress and the skin scrape cortisol peak has only been studied in bottlenose dolphins (time lag 45–60 days) [41]. Such information would be crucial before using skin scrape as a material in future studies of cetacean stress, and is vital to identifying the cause of stress. In the current study, the time lag in the beluga whales’ stress response was found to be 68–72 days, which is similar to the estimates of the epidermal turnover rate, which ranges from 70 to 75 days in beluga whales; this can be determined by incorporating [6-^3^H] thymidine into the nuclei of the cells that synthesize DNA [50]. The longer time lag in beluga whales than in bottlenose dolphins may be due to the thickness of the beluga epidermis and the intrinsic rate of growth of basal cells, which leads to a longer wound healing time [51]. It should be noted that the time lag estimation would be more precise by employing an out-of-water stress test [13,41] or administering hydrocortisone [42], which was not prohibited in the current study. Nonetheless, the estimated time lag in the stress response of beluga whales provides an opportunity for a retrospective study and an evaluation of the stress and welfare of cetaceans in order to discover the underlying acute or chronic stress that is unnoticed by traditional observation. In this study, for example, the food intakes were successful for these animals and no negative behaviors (e.g., increased respiration rate, prolonged floating time, increased frequency of fast swimming, fluke slapping or jumping) were observed during all the event periods, except for event IV (enteritis in beluga C). This showed that the stress experienced by captive beluga whales might not be evaluated effectively by clinical performance.

The extraction efficiency (41.6% ± 15.9%) in this study is lower than that in a previous study (68.5% ± 13.9%) [39] using a similar extraction protocol in order to homogenize the skin blubber samples. The possible explanation for this is that the lipid content is inversely proportional to the duration of exposure to seawater [52]. After collecting scrape samples in the current study, the removal of seawater was performed immediately. However, it was difficult to completely remove the water after sample collection as the samples were taken from the outermost part of the skin and were in a liquid–solid form. The skin scrape cortisol concentrations presented in the current study were comparable to those found in previous studies, which were 0.6–15 ng/g (dry weight, dw) in bycaught *P. phocoena* [46] and 0–8.4 ng/g (dw) in the captive *T. truncatus* [41]. Meanwhile, the minor changes in the cortisol concentration could be detected via EIA, although comparatively low concentrations (median value: 1.14 ng/g, wt) were detected in the scrape samples. This provided evidence to support the suggestion that the EIA could be used to monitor the dynamic changes in the scrape cortisol concentration. No differences in the blubber cortisol concentration across the dorsal, lateral, and ventral around the whales’ girth were reported [53]. In the current study, it was shown that no obvious differences in the cortisol concentration were found between the scrape samples taken from the flipper and the fluke (Figure 5). This indicated that trainers would have the flexibility to collect samples from either the flippers or fluke on the same day, and even pool the samples, since it may take nearly one week for enough scrape to be renewed for cortisol measurement. This would make it possible to maintain routine sampling so that a baseline can be established in order to evaluate the stress.

The three whales had been kept in the same captive environment. Therefore, it was assumed that they were all affected by the same stressors. However, their cortisol concentrations varied during the same events (Figure 6). A possible explanation for this is different stress responses among individuals. The comparatively high cortisol concentration in some samples and the higher CV in beluga B indicated that this whale likely showed a stronger response to certain events than the other two whales. According to the trainers’ opinion, which was not based on a statistical analysis, beluga B was more sensitive to environmental changes. In additional, beluga C showed some high cortisol concentrations during some events. However, when the outliers were not included, the CV of beluga C decreased by 70% and was the lowest CV among the three belugas. This showed that beluga C might have only responded to significant stressors. Beluga A was considered to be the stable one by the trainers. It showed only one outlier and its CV differed by only 23% when the outliers were not included, suggesting that this whale was relatively stable compared to the other two animals. Individual differences regarding the cortisol concentration were observed among these three whales, indicating that the whales’ characteristics could be observed in the scrape cortisol concentration changes. It was suggested that the scrape cortisol analysis could potentially be used in individual evaluation rather than in group evaluation.

The enteritis experienced in event IV was considered to have had a progressive effect on the animal’s behavior and to have gradually affected the vocal behavior. On the day with the lowest number of calls (4th April), the trainer and veterinarian diagnosed the animal with enteritis, and it was suggested that beluga C was seriously discomforted on that day. The distribution pattern of calls on 4th April, 5th and 6th was similar (Figure 8), indicating that this was a big stressor and had a long effect on the acoustic behavior. This showed that acoustic behavior may be a non-invasive tool that can be used to monitor the progress of internal illness. In the current study, it remained unknown as to whether the decrease in the total number of calls was due to the decrease in the vocal behavior of a single ill whale or all three whales. Many odontocete species present complex social behaviors that help them to cope with stressors, maintaining individual survival and reproductive success [3]. It was hypothesized that the social cohesiveness of this whale group may lead to a group response when one individual is stressed, which requires further study.

The number of calls obviously decreased on the day of the selected events (Figure 8). A similar finding was reported in a previous study on captive beluga whales [28]. Due to the longer recovery time in events I (maintenance) and IX (impulsive sound), it was suggested that these two events caused more stress compared to events VII (repairment) and VIII (fire drill). This indicated that responses to different stressors and the consequent recovery time are not uniform. The noise and vibration from the filtering system maintenance and the impulsive noise from the sea embankment may have been considerable stressors to the captive beluga whales. In events I, IV, and IX, the pulse call were in the vast majority on the day of the event compared to the days before the event. The phenomenon was not observed in events VII and VIII. It was supposed that a specific call distribution and change pattern, accompanied by a paralleled physiological parameter profile, is for a certain stressor; this warrants further investigation. It is intriguing that the trainers did not observe any negative behavior during all the event periods, except for event IV, but that the skin cortisol and acoustic behavior analysis told a different story. The results provided supporting evidence that acoustic behavior monitoring would help identify and mitigate the underlying stressors and improve the husbandry of captive cetaceans.

## 5. Conclusions

Simultaneously analyzing behavioral and physiological parameters can thus be considered as an effective method with which to evaluate animal welfare accurately [28]. An approach that utilizes cortisol and specific behavioral monitoring could be helpful in representing the diverse and dynamic aspects of welfare that are experienced by animals [54]. In terrestrial mammals, concurrently assessing stress and welfare by using physiological and behavioral parameters has been reported [54,55,56,57,58]. The evaluation could be enhanced if individual details were considered [59]. Here, we have presented the first study on the time lag between skin scrape cortisol concentrations and stressful events, and have evaluated two indicators in order to identify and analyze stress in captive beluga whales at the individual (cortisol) and group (acoustic) levels. With well-trained personnel and proper equipment that can efficiently collect and analyze data, this method would substantially contribute to cetacean husbandry. This study concludes with implications for future studies regarding the identification of potential sources of stress by using non-invasive vocalization analysis and the development of strategies to reduce or eliminate it. The findings of the current study provide fundamental information for the future study of stress and welfare science regarding captive cetaceans, which can subsequently improve their quality of life. It is recommended that aquaria may consider incorporating these indicators in order to monitor stress and animal welfare. It may also help us to incorporate knowledge into adaptive management schemes during coastal construction and other human activities.

## Figures and Tables

**Figure 1 animals-13-01521-f001:**
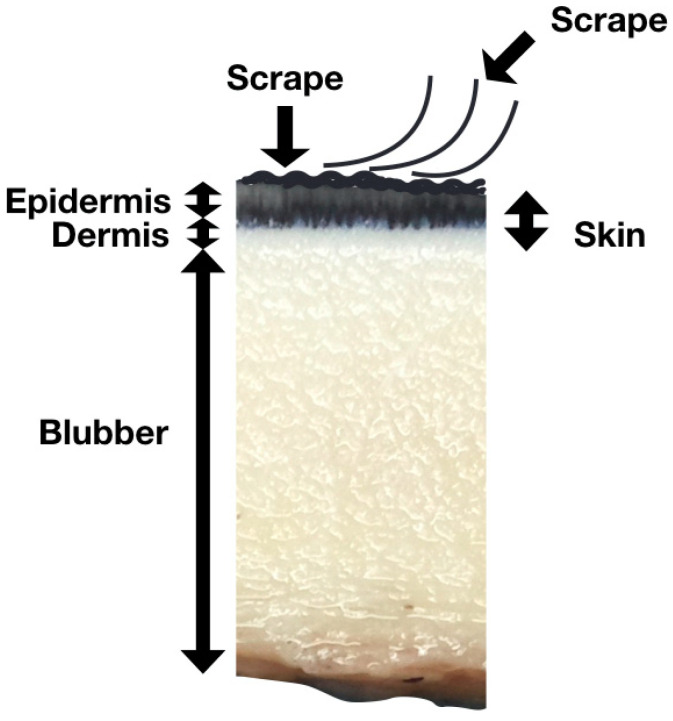
Cross-section of cetacean skin. The skin scrape comprises the outermost, sloughed and superficial dead layers of the skin.

**Figure 2 animals-13-01521-f002:**
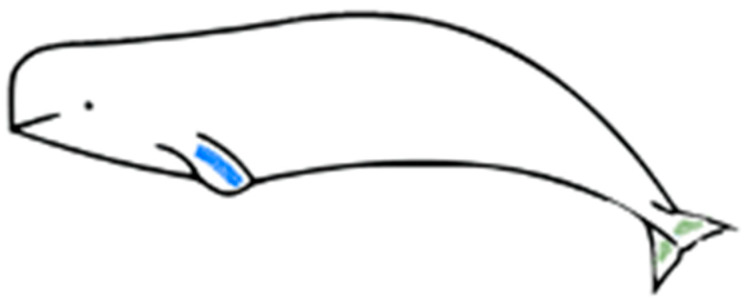
Sloughed skin scrape sampling location during the routine monitoring of the beluga whales. Blue: sampling location of beluga B and C. Green: sampling location of beluga A.

**Figure 3 animals-13-01521-f003:**
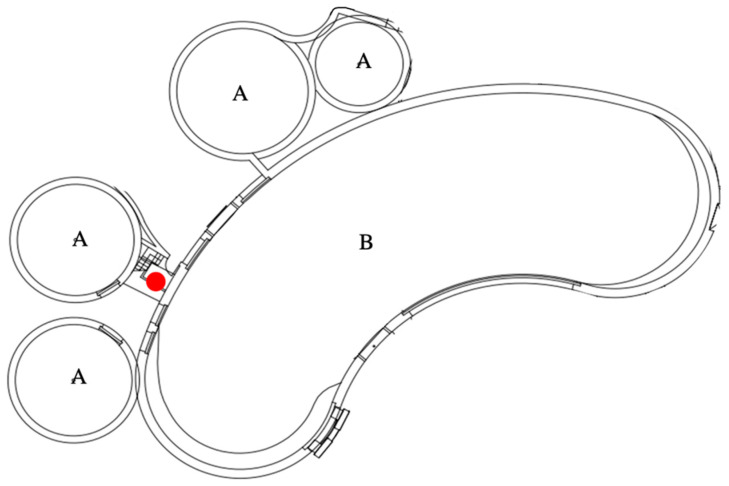
The floor plan of the aquarium. A and B are the pools in which the belugas live. The red spot was where the SM4 and hydrophone were placed.

**Figure 4 animals-13-01521-f004:**
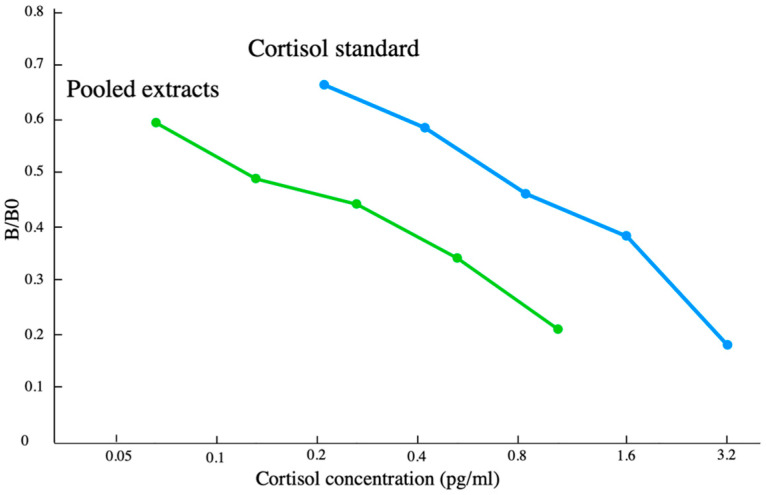
Results from linearity assessment of EIA standard with scraped extracts.

**Figure 5 animals-13-01521-f005:**
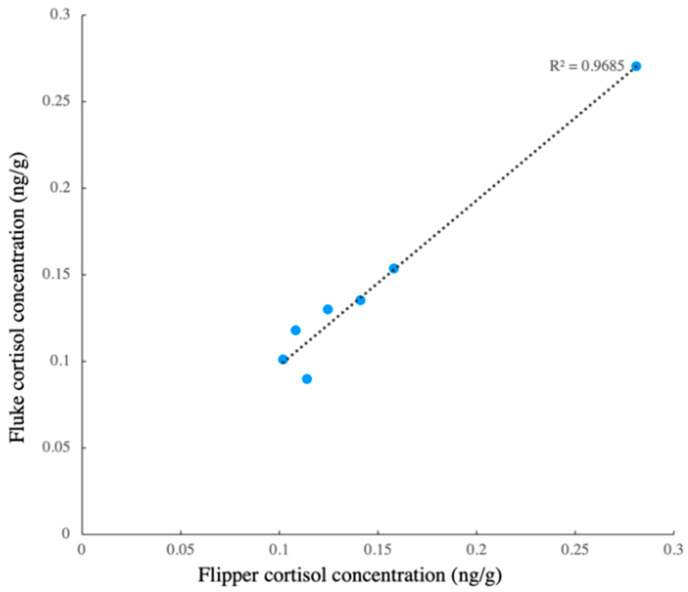
Linear regression of flipper–fluke cortisol concentration.

**Figure 6 animals-13-01521-f006:**
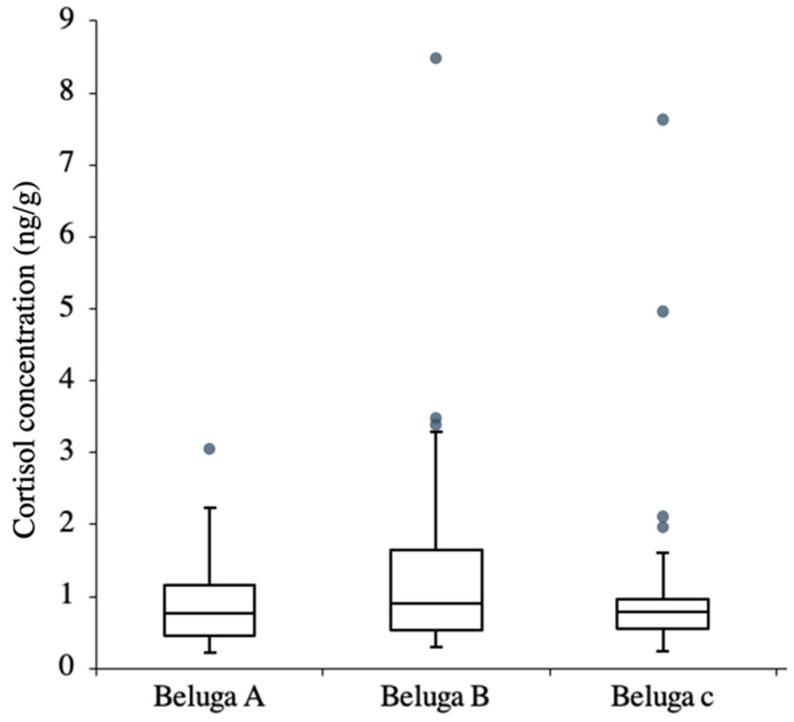
Box plot of scrape cortisol concentration. Grey dots are the outliers.

**Figure 7 animals-13-01521-f007:**
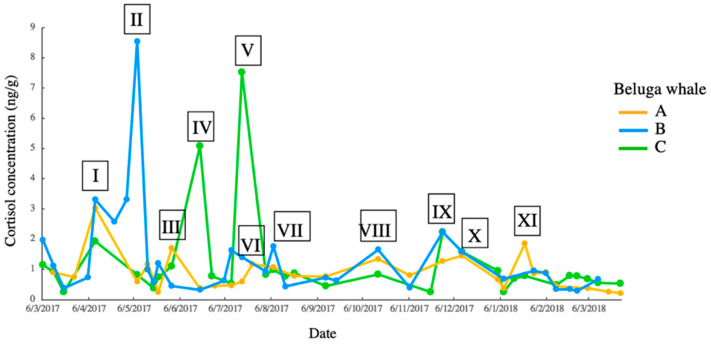
Cortisol concentration–time graph and corresponding events. Event I to XI were identified as events by matching the peak of the cortisol concentration with the date of the event.

**Figure 8 animals-13-01521-f008:**
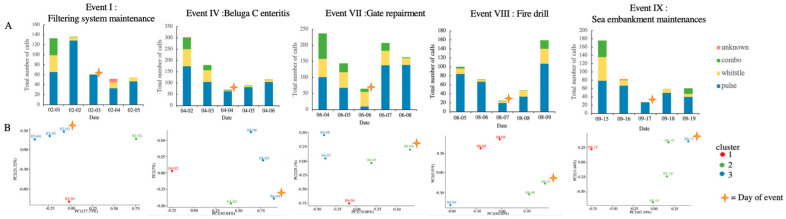
The analysis of the number of 5-day calls in each selected event. (**A**) The stacked bar chart of different calls. (**B**) The PCA analysis result. PC1 and PC2 ranged between 74.08% to 94.20% and 25.10% to 5.55%, respectively. A star indicates the day of the event.

**Table 1 animals-13-01521-t001:** Time lag estimation of cortisol signal in scrape.

Day	Beluga Whale			Points
	A	B	C	
28–32	3	2	2	7
38–42	4	4	2	10
48–52	5	2	3	10
58–62	5	3	2	10
68–72	5	5	4	14
78–82	4	3	2	9

**Table 2 animals-13-01521-t002:** Description of the events according to husbandry and medical records.

		Environmental Changes	Physiological Changes

I		Filtering system maintenance	/
II		Filtering system maintenance	/
III		Minor maintenance	/
IV		/	Beluga C enteritis
V		Construction of false bottom in medical pool	/
VI		Gate repairment	/
VII		Gate repairment	/
VIII		Fire drill	/
IX		Impulsive sound of unknown-source from the sea embankment	/
X		Sound exposure experiment on beluga A	/
XI		Maintenance of false bottom	

## Data Availability

Data is contained within the article.

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
