# Peer review of "Skin Cortisol and Acoustic Activity: Potential Tools to Evaluate Stress and Welfare in Captive Cetaceans"

_animals, 2023, doi:10.3390/ani13091521_

Round 1

Author Response

We would like to take this opportunity to express our sincere thanks to you who identified areas of our manuscript that needed corrections or modification. We would like also to thank you for allowing us to resubmit a revised copy of the manuscript.

Title After reading the manuscript, I feel that the title should be altered, as currently it reads in a way that suggests you are combining cortisol results and acoustics into one tool to evaluate welfare. However, no such work is undertaken as you examine these separately. Please alter the title to reflect this i.e. potential tools

This has now been amended (line 2).

Abstract

Lines 11-12: You mention animal stress here, but it reads as if discussing both stress as a mental state as it may be perceived by the public, and stress as in the physiological state. I believe that careful consideration of word choice should be made here to ensure that you are only discussing stress as the physiological state.

This has now been amended (line 13).

Line 17: Over what time frame were acoustic recordings made?

This has now been amended (line 16).

19: Skin scrapes cannot be non-invasive. So, either remove that wording, or if you are talking about sloughed skin that was not removed by humans then please alter the text to reflect that.

This has now been amended (line 19).

Line 21 provide the results

This has now been amended (line 21-23).

Line 25: You have separated out animal welfare and psychological well-being. What approach/conceptualisation of animal welfare are you using here? This final sentence also suggests that you are discussing stress in terms of psychology rather than as a physiological state, which is not correct from an animal welfare science view.

This has now been amended (line 27).

Introduction The first paragraph of the introduction seems to be a jumble of mixed ideas that do not really follow on from each other. I am uncertain what the authors are really trying to say in this paragraph. Line 29-30: You are mixing up two separate formulations of animal welfare. The Five Domains Model does not include the Five Freedoms. Please decide which of these two approaches you are using and mention these specifically, alternatively if you are taking ideas from both please provide details on each as they are not the same thing and have different ways of assessing welfare and the outcomes of such assessments.

This has now been amended (line 32-33).

Line 36: I do not understand this sentence or its purpose

This has now been amended (line 38-40).

Line 40: Stressors will disrupt homeostasis, but these will not solely be negative experiences that therefore impair animal welfare as suggested.

This has now been amended (line 44).

Line 42: I do not disagree with this point, but it seems to have been randomly added on to the end here with no clear story leading to this point. This whole first paragraph needs to be considerably reworked.

Line 33-35 also mentioned captive animals

Line 43: Do you mean stress response?

This has now been amended (line 46).

Line 44: Stress response cause physical and physiological changes, not emotional. There can be alterations to affective states experienced due to physiological stress but this is not part of stress response. There is much scientific validation to be done on the various physical, behavioural and physiological responses that occur to understand how they may infer affective states.

This has now been amended (line 47).

Line 46-48: Please include references

This has now been amended (line 51).

Line 48: It can also be through prolonged duration of stress response not only increased frequency. Line 49: Please change would to can

This has now been amended (line 51).

Lines 49-50: You need to introduce the context of captive animal welfare somewhere. You have not provided any kind of context/explanation of captive animal situation, why it is important to assess physiological stress and why we have any kind of responsibility to their welfare.

Line 33-35 and 45 also mentioned captive animals.

 Line 69- breaks the flow of the text.

This has now been amended (line 70-73).

 Lines 73-85: There seems to be a lot of unnecessary detail here that is not clearly linked to the aims of the current study. This may be ok to keep once a paragraph of context for the present study is included but currently it appears random.

Since it was described that the empirical studies are a few (line 72-76), the details maybe help readers to know the brief history and appreciate the work in this study.

Line 75: Remove a before construction

This has now been amended (line 78).

Line 76: This suggests that the previous study was not assessing physiological stress as part of a welfare assessment?

This has now been amended (line 79).

Line 88: Are these piglets those referred to in the previous sentence regarding castration?

This has now been amended (line 91).

Line 89: Again, this comes put of nowhere quite randomly. You have not yet introduced belugas in

This has now been amended (line 89-101)

 Line 90: Please change to vocal repertoire- please ensure this is altered throughout the text

This has now been amended (line 91)

Line 90: Generally, once the scientific name has been provided at the first mention, the common name should be used throughout the text.

This has now been amended

Line 94: Please change different to various

This has now been amended (line 99)

Line 95-100: What study are you talking about? Where is the context for this study and what is was assessing when examining vocal repertoire?

This has now been amended (99-101)

Line 101: Please alter the sentence to read Vocalization parameters alone, such as vocal rate, intensity, and frequency range, have been found to vary with stress.

This has now been amended (Line 102)

 Lines 107-115: What is the context for all of this information? Will you be assessing cytokines as part of the analysis? This is a lot of detail for a method that you do not use.

[37] and [38] demonstrated what we mentioned in line 117-118.

some examples and references Lines 120-124: I am still unclear what the aims of this study are or what will be done? Please provide some detailed, specific aims.

This study evaluated the non-invasive vocalization and sloughed skin scrape cortisol analysis specific associated to potential stressful contexts as tools for identifying the sources of stress in captive beluga whales.

 Methods Line 128: I am highly concerned about the animal ethics and permits allowing the capture of these animals and note that the authors have not provided any details on ethics or permits. This alone makes me question whether such a study can be published.

The permit is showed Line 140-141

Line 133: What was included in the physiological and health assessments?

This has now been amended (Line 135-137)

Line 144: What type of tongue depressor was used? Please provide additional description or an image. suggesting they may not have been at times.

This has now been amended (Line 146)

Line 150: How were the skin samples kept prior to centrifuge?

This has now been amended (Line 152-153)

 Line 153: Please provide a more detailed description of the figure as it does not only show cetacean skin. It is useful if a figure caption provides all information without requiring the reader to go back to the text.

This has now been amended (Line 155-156)

Line 177: Please alter to studies

This has now been amended (Line 180)

 Line 201: Please change to and/or

This has now been amended (Line 203)

Line 210: I am uncertain where these various time lag in days have come from? Can you provide references to support why you assumed that a certain cortisol response was linked to a particular event?

In Discussion the first paragraph mentioned the previous studies showed the time lag in bottlenose dolphin and beluga is 45-60 days and 70-75 days, respectively.

Line 216-221: This is very difficult to follow. Please re-write to better explain how this was calculated a significant amount of guesswork around what a cortisol reading relates to?

This has now been amended (Line 218-223)

Line 226: Was this recording made daily over the course of the 1 year period?

Yes (line 231)

Line 227: Why was within half an hour of a husbandry procedure considered sufficient time to not effect acoustic activity? Please provide reference

This has now been amended (Line 227-230)

Line 229-233: This seems to be related to data analysis in terms of how cortisol results and acoustics were examined synchronously? If so, please move this to a separate section of data analysis

This has now been amended (Line 231-235)

Line 231: Do you mean audially here? i.e. by sound

Aurally and visually, by hearing and observing (line 234)

Line 245: What were these statistical tests used for? To test/ show what?

Just like it is mentioned Line 245-250, for cortisol and acoustic activities analyses (line 122-124).

Line 248: To do what? What were results of PCA used to show?

Fig. 8 showed the call distribution differed among the five days of every event based on PCA (line 308-310)

Section 2.7 at Line 242: How do these tests link to the aims of the study and test such aims/hypotheses?

Just like it is mentioned Line 245-250, for cortisol and acoustic activities analyses (line 122-124).

 Results Lines 251-253: This should be included in methods not results

This showed the extraction efficiency result. The methods for extraction efficiency is mentioned Line 18.

Line 264: Please move to methods

This has now been amended (Line 246, 266-268)

Line 280: I cannot see any scientific justification for how the authors have decided that a particular change in cortisol concentration relates to an event.

Line 209-211 mentioned there is a time lag between the dates of event and response.

Lines 285-293: This should all be included in methods, these are not results

The selected events were selected based on the husbandry and medical record and recording quality (Line 231-233)

Table 2 Line 295: Please provide a more detailed Table caption

This has now been amended (Line 295)

 Figure 7 Line 296: Please provide a larger, high resolution figure as the labels are not easily readable

Yes, I will cooperate with editors during the proofreading.

Lines 300-302: This should be included in methods

The selected events were selected based on the husbandry and medical record and recording quality (Line 231-233)

Line 304: Please include n= in the parentheses of the number of different call types.

This has now been amended (Line 307)

Line 305 306-alter this sentence

This has now been amended (Line 308-310)

Lines 307-310: Please re-write these results, it is not clear what the authors are trying to say. Also please provide details on the results of the PCA, what percentage of variation was explained by PC1 and PC2? And how does this relate to the results of types of vocalisations on various days?

This has now been amended (Line 316-317). The PCA showed that the call distribution differed among the five days of event. The number of calls plummeted on the day of the event, which was discussed the importance of this finding in Discussion.

Figure 8 Line 311: Please provide a detailed figure caption. It is not clear what the figure shows e.g. what is the star on the stacked bar plot?

This has now been amended (Line 317)

I will cooperate with editors during the proofreading to improve the resolution of figure and the star.

Discussion

Please begin your discussion with a small paragraph that synthesises the overall findings of your study. Due to the lack of aims/hypotheses provided, it is not clear what the study was trying to achieve and therefore whether the aims were achieved.

This study evaluated the non-invasive vocalization and sloughed skin scrape cortisol analysis specific associated to potential stressful contexts as tools for identifying the sources of stress in captive beluga whales. (Line 122-124)

Line 325: Whilst skin scraping could be considered less invasive that blood sampling for example, I do not believe it should be considered non-invasive. Please alter this where relevant.

This has now been amended (Line 155, 158, 329)

Line 330: A word seem to be missing

This has now been amended (Line 335)

Line 331-333: So is this one of the aims of the current study? To provide evidence of the potential time lag between a stressor occurring and the resulting cortisol response in the skin of Beluga? If so, this needs to be made much clearer throughout the manuscript as I am only coming across this as a possible aim part way through the discussion.

Line 208, 281-283 showed that time lag estimation is one part of the cortisol detection.

 Line 339: This is the first mention of behaviour (non-acoustic) in this manuscript. You cannot bring in such points at the end of a manuscript when they have not been presented or analysed. Please either remove any such information or provide the information if available as applicable throughout the manuscript.

This has now been amended (Line 295-296)

Line 341: You cannot make this assumption without providing further data to evidence this point.

This has now been amended (Line 346)

 Line 361: This point is something that had concerned me in the methods earlier and how you have used the results of cortisol to suggest a particular stressor. If you have only collected these samples once per week, and this is the frequency they can be collected at, how can you be sure of the stressor that caused any alteration to cortisol, since a variety of stressors may occur each day for many in a row.

It is true that this study is more like a retrospective study. We carefully analyzed the husbandry and medical record, and so we knew when the presumed “effective” events happened.

Line 362: Due to the point raised above, I do not think that stating fine scale is appropriate. This would be weekly sampling versus behavioural analyses that could be conducted per minute.

This has now been amended (Line 366)

Line 367 trainers’ opinions: this has been provided. If such discussion is going to be utilised such data should be provided including how this was collected and preferably using verbatim transcription of such qualitative data. This would provide more depth to the current data and would allow for improved interpretation of results.

Trainers’ opinion is a subjective judgement, not based on scientific or systematic analyses. We did a comparison between subjective (trainers) and objective (cortisol) findings.

Line 393: What event? Do you mean any of the stressors being present? Does this mean that you also carried out such analysis on all other days apart from those chosen to examine for changes in vocalisation depending on any stressor being present?

This has now been amended (Line 397)

Lines 395-396: Please provide the description of the event rather than the number, as written it is difficult to follow what stressor appears to be more impactful.

This has now been amended (Line 399-401)

Line 403: Please provide information on the behavioural monitoring conducted and the results if using such comparisons.

This has now been amended (Line 135, 296)

Lines 408-409: This is true, however, in this study you have not provided an analysis of these two types of measure alongside each other. Please consider adding a section in the methods and results to enable an examination of these two indicators concurrently, so that you can then discuss this in detail.

This has now been amended (Line 413-421)

Line 416: Until this sentence it has not been clear throughout the manuscript how these two different measures were being applied. Notably, you should be careful suggesting that you are simultaneously assessing welfare using two parameters that are being measured at different scales i.e. one at the individual level and the other at the group level.

This has now been amended (Line 419-421)

Reviewer 2 Report

The manuscript is interesting and the method to evaluate the welfare condition in captive cetaceans could be useful if well-validated, however, I have some concerns about the methodology used to estimate the time lag between a stress event and the increase in cortisol level in skin scrape samples. Data demonstrate that there is a local production of glucocorticoids at the skin level that, of course, it is not correlated with the circulating glucocorticoid levels due to HPA axis activation following a stress stimulus. Consequently, correct identification of the lag time is fundamental to can use this tool in welfare studies and reduce the eventual effects of local hormone production. Maybe the injection of ACTH could give a more consistent and reliable identification of the correct response time.

Minor points

The figure 7 and figure 8 seem to have different date. Figure 8 is really too small and difficult to be read.  

Author Response

We would like to take this opportunity to express our sincere thanks to you who identified areas of our manuscript that needed corrections or modification. We would like also to thank you for allowing us to resubmit a revised copy of the manuscript.

For Time Lag Estimation, we amended the paragraph (line 218-223)

Because ACTH injection experiment was not allowed by the aquarium and government, it is true that this study is more like a retrospective study. We carefully analyzed the husbandry and medical record, and so we knew when the presumed “effective” events happened.

Line 300, 303 315 
the date of cortisol concentration and the date of events are different because of time lag. 

We will cooperate with editors during the proofreading to improve the resolution of Figure 8.

Round 2

Reviewer 1 Report

Review 2 of: Skin cortisol and acoustic activity: a potential tool to evaluate stress and welfare in captive cetaceans

Line 19: These are not non-invasive skin samples, please remove this word as it is not accurate to the method.

The first paragraph of the introduction remains confusing; I believe this may be due in part to a need to improve the English language and style. I recommend that the authors use native English review/editing proof readers.

Line 60-61: Please change ‘would be’ to ‘are’

Line 63: Please remove ‘maintenance’

Line 74: Please alter this sentence ‘Using different assessments to identify stressors could be interpreted as a more valid indicator of stress than GCs alone.’ To read: ‘Using multiple measures to identify stressors could enhance the scientific validity of GCs as indicators of welfare’.

Lines 77-80: These two sentences should be better combined to improve reading, something like: ‘Despite the fact that cetaceans have been maintained in a captive environment for over 150 years, limited scientific research has been dedicated to their welfare [24/25]. Indeed, there are only several established methods to quantify and address captive cetacean stress and welfare [24,25’.

Lines 91-93: Please alter the construction of this sentence to improve the English: ‘One study found that behavior and salivary cortisol concentration varied among dolphins in open and closed facilities, suggesting that the design of pools may influence welfare [27]’.

Line 105: Remove this sentence, it is repeated below and is not well placed here.

Line 114: I do not believe that you can say ‘more accurate’ when there are so few studies. Please alter this sentence to something more appropriate e.g., ‘Acoustic activities have been shown to be sensitive to environmental stressors’

Line 118: in what species?

Line 131: Remove the word ‘previous’

Line 137-140: This is not clear and requires significant English improvements. Please carefully state what the aims are, here you are not describing what you plan to achieve only what you are doing.

Line 142: What is ‘it’?

Line 146: I am still very concerned about the ethics of how these animals were captured. Details should be provided on the ethics and permits for the capture of such animals.

Line 151-152: If you are suggesting that careful consideration of husbandry records were used to ensure that the only stressor to these animals were those tested then you must provide detail on how such assessments were carried out, which parameters were examined and how were these data used to inform the use of data in your study.

Line 226 section 2.5: It is still not clear from this section how it was decided upon that a certain stress response was linked to a particular stressor event. You are stating that the number of days that the response could occur is anywhere between 28 and 82 days. You have ‘guessed’ and used the data to fit your hypothesis of a particular stressor event affecting the parameter. I do not believe that this is a valid way to be assessing such a parameter. You may be correct in that the cortisol is affected by the particular stressor, but this is not a scientifically valid method. I believe that the use of such a method without additional evidence could harm welfare science as you are not considering alternative hypotheses. A good example of the scientific method being used for this type of work is reference 41. I suggest if using this type of data you must provide much improved detail and data on the methods and analysis.

Line 251: Based on what in the husbandry and medical records? What were the ‘several event days’? How were such event days chosen and why? As written this sounds like the data has been manipulated to fit the hypothesis being tested.

Line 264 section 2.7: This provides no information about how analyses were carried out. You have stated that you did a statistical test for certain data, but why, what were you testing, what correlations were being examined for, how were these used to test your hypothesis? These should be detailed and specific so that someone else reading your paper could replicate what you have done.

Lines 302-306: As written this is an example of the data being fitted to the hypothesis rather than evidence to support it. I believe that these methods need to be re-evaluated and data analysis improved.

Lines 333-336: This is very difficult to understand. I believe that a combination of providing additional information describing the results and improvements in the English language and style would help here.

Line 337 Figure 8: These figures are completely illegible. I cannot comment on their usefulness/appropriateness.

Line 342: You should begin your discussion with a short paragraph to describe the overall key results of your study and why these are important before you begin discussing the literature. As an example, you don’t mention any results of your own study until line 359.

Line 362: Please remove ‘which was estimated in beluga whales’ as you say this above.

Line 369: What were considered to be ‘negative behaviours’? No mention of these was provided earlier.

Lines 389-390: Based on this it seems that you would not be able to collect skin samples regularly enough to analyse stressor events to a particular day.

Line 396: How did trainers evaluate this? As mentioned in the last review you need to provide data, including how this was collected and preferably using verbatim transcription of such qualitative data. This would provide more depth to the current data and would allow for improved interpretation of results.

Line 409: Since this animal had a known health issue you cannot include it in the analysis to suggest that the environmental stressor was the cause of any change in welfare indicators.

Lines 429-433: This needs to be checked for English. It is currently confusing as written. Do you mean that the frequency or relative duration increased? Are you suggesting that on these same days/events the physiological parameter i.e., skin cortisol also showed changes? If so, these results comparing the two data types have not been presented comparatively.

Line 433: Again here you discuss trainer observations, but you have provided no methods on what they assessed or how, and how these data were utilised.

Line 438: Please change back to ‘Simultaneously’

Line 439: You cannot measure welfare. Welfare is inferred by observing and/or measuring indicators. Please alter this sentence appropriately.

Author Response

We would like to take this opportunity to express our sincere thanks to the reviewer who identified areas of our manuscript that needed corrections or modification. 

Line 19: These are not non-invasive skin samples, please remove this word as it is not accurate to the method.

We think the sample collection method is extremely similar to that in Bechshoft et al. 2020. It was mentioned in Bechshoft et al. 2020 : “This study develops and validates a non-invasive method for measuring cortisol and other hormones related to stress, health and reproduction in the skin of cetaceans.”  

The first paragraph of the introduction remains confusing; I believe this may be due in part to a need to improve the English language and style. I recommend that the authors use native English review/editing proof readers.

Yes, the English editing will be performed after the content is matching reviewer’s comments.

Line 60-61: Please change ‘would be’ to ‘are’

This has now been amended (Line 61)

Line 63: Please remove ‘maintenance’

This has now been amended (Line 63)

Line 74: Please alter this sentence ‘Using different assessments to identify stressors could be interpreted as a more valid indicator of stress than GCs alone.’ To read: ‘Using multiple measures to identify stressors could enhance the scientific validity of GCs as indicators of welfare’.

This has now been amended (Line 74-76 )

Lines 77-80: These two sentences should be better combined to improve reading, something like: ‘Despite the fact that cetaceans have been maintained in a captive environment for over 150 years, limited scientific research has been dedicated to their welfare [24/25]. Indeed, there are only several established methods to quantify and address captive cetacean stress and welfare [24,25’.

This has now been amended (Line 78-81 )

Lines 91-93: Please alter the construction of this sentence to improve the English: ‘One study found that behavior and salivary cortisol concentration varied among dolphins in open and closed facilities, suggesting that the design of pools may influence welfare [27]’.

This has now been amended (Line 96-97 )

Line 105: Remove this sentence, it is repeated below and is not well placed here.

This has now been amended (Line 111 )

Line 114: I do not believe that you can say ‘more accurate’ when there are so few studies. Please alter this sentence to something more appropriate e.g., ‘Acoustic activities have been shown to be sensitive to environmental stressors’

It was the suggestion of the authors in [28] not us, and we used “supposed”.

Line 118: in what species?

This has now been amended (Line 122 )

Line 131: Remove the word ‘previous’

This has now been amended (Line 135 )

Line 137-140: This is not clear and requires significant English improvements. Please carefully state what the aims are, here you are not describing what you plan to achieve only what you are doing.

This has now been amended (Line 142-150 )

Line 142: What is ‘it’?

This has now been amended (Line 152 )

Line 146: I am still very concerned about the ethics of how these animals were captured. Details should be provided on the ethics and permits for the capture of such animals.

This has now been amended (Line 157-158)

Line 151-152: If you are suggesting that careful consideration of husbandry records were used to ensure that the only stressor to these animals were those tested then you must provide detail on how such assessments were carried out, which parameters were examined and how were these data used to inform the use of data in your study.

This has now been amended (Line 162-168 )

Line 226 section 2.5: It is still not clear from this section how it was decided upon that a certain stress response was linked to a particular stressor event. You are stating that the number of days that the response could occur is anywhere between 28 and 82 days. You have ‘guessed’ and used the data to fit your hypothesis of a particular stressor event affecting the parameter. I do not believe that this is a valid way to be assessing such a parameter. You may be correct in that the cortisol is affected by the particular stressor, but this is not a scientifically valid method. I believe that the use of such a method without additional evidence could harm welfare science as you are not considering alternative hypotheses. A good example of the scientific method being used for this type of work is reference 41. I suggest if using this type of data you must provide much improved detail and data on the methods and analysis.

This has now been amended (Line 241-257, 380-383)

We knew the time lag estimation could be better by using other ways (stress test or administering drugs), but we were not allowed to do those. Hopefully someone can do it on beluga whales in the future.

Line 251: Based on what in the husbandry and medical records? What were the ‘several event days’? How were such event days chosen and why? As written this sounds like the data has been manipulated to fit the hypothesis being tested.

This has now been amended (Line 266-267, 322-324)

Line 264 section 2.7: This provides no information about how analyses were carried out. You have stated that you did a statistical test for certain data, but why, what were you testing, what correlations were being examined for, how were these used to test your hypothesis? These should be detailed and specific so that someone else reading your paper could replicate what you have done.

We described the statistical tests followed by the corresponding objectives in M&M and Results.

Lines 302-306: As written this is an example of the data being fitted to the hypothesis rather than evidence to support it. I believe that these methods need to be re-evaluated and data analysis improved.

We obtained the number and date of the responses (line 247-250), and knew that six responses in beluga A, five in beluga B and four in beluga C (line 319-320). Since we got time lag estimation (line 317) through the analysis (line 251-257), 11 events were the potential causes of the responses (line 320-321).

Lines 333-336: This is very difficult to understand. I believe that a combination of providing additional information describing the results and improvements in the English language and style would help here.

Yes, we will cooperate with the editors to improve the typesetting of Figure 8 and do English editing during proofreading stage.

Line 337 Figure 8: These figures are completely illegible. I cannot comment on their usefulness/appropriateness.

Yes, we will cooperate with the editors to improve the typesetting of Figure 8.

Line 342: You should begin your discussion with a short paragraph to describe the overall key results of your study and why these are important before you begin discussing the literature. As an example, you don’t mention any results of your own study until line 359.

This paragraph focused on the discussion on time lag estimation only.

Line 362: Please remove ‘which was estimated in beluga whales’ as you say this above.

This has now been amended (Line 377-378)

Line 369: What were considered to be ‘negative behaviours’? No mention of these was provided earlier.

This has now been amended (Line 387-388)

Lines 389-390: Based on this it seems that you would not be able to collect skin samples regularly enough to analyse stressor events to a particular day.

We said the trainers have the flexibility to collect samples from either flippers or fluke, so the frequency of collection could be increased.

Line 396: How did trainers evaluate this? As mentioned in the last review you need to provide data, including how this was collected and preferably using verbatim transcription of such qualitative data. This would provide more depth to the current data and would allow for improved interpretation of results.

This has now been amended (Line 418-420)

Line 409: Since this animal had a known health issue you cannot include it in the analysis to suggest that the environmental stressor was the cause of any change in welfare indicators.

Health issue is an intrinsic stressor (Line 25, 341-342)

Lines 429-433: This needs to be checked for English. It is currently confusing as written. Do you mean that the frequency or relative duration increased? Are you suggesting that on these same days/events the physiological parameter i.e., skin cortisol also showed changes? If so, these results comparing the two data types have not been presented comparatively.

This has now been amended (Line 450-452).

The estimated time lag is a piece of time (68-72 days) so skin cortisol showed changes but not on “that” day.

Line 433: Again here you discuss trainer observations, but you have provided no methods on what they assessed or how, and how these data were utilised.

This has now been amended (Line 387-388).

Line 438: Please change back to ‘Simultaneously’

This has now been amended (Line 460).

Line 439: You cannot measure welfare. Welfare is inferred by observing and/or measuring indicators. Please alter this sentence appropriately.

This has now been amended (Line 461).

Reviewer 2 Report

I suggest an english language revision of the manuscript to increase readability. 

Author Response

We would like to take this opportunity to express our sincere thanks to the reviewer who identified areas of our manuscript that needed English language  revision. We will send the revised manuscript for English editing when the content is matching all reviewers' comments. 

Sincerely,

Wei-Cheng Yang